# The Effect of Medical Choice on Health Costs of Middle-Aged and Elderly Patients with Chronic Disease: Based on Principal-Agent Theory

**DOI:** 10.3390/ijerph19137570

**Published:** 2022-06-21

**Authors:** Dongxu Li, Min Su, Xi Guo, Weile Zhang, Tianjiao Zhang

**Affiliations:** School of Public Administration, Inner Mongolia University, Hohhot 010070, China; 22010007@mail.imu.edu.cn (D.L.); 13327118080@163.com (X.G.); zhangweile78@163.com (W.Z.); ztj13948536101@163.com (T.Z.)

**Keywords:** chronic disease patients, medical choice, health costs, quasi-natural experiment, principal-agent

## Abstract

(1) Background: The discussion on how to reduce the health costs of chronic disease patients has become an important public health issue. Limited research has been conducted on how chronic disease patients’ medical choice of public and private medical institutions affect health costs. (2) Methods: This study used the panel data composed of the China Health and Retirement Longitudinal Survey (CHARLS) from 2011 to 2018, adopted the quasi-natural experimental research method, and set up a control group and an experimental group that chose public medical institutions and private medical institutions, to analyze the association between the medical choice and health costs of chronic disease patients. (3) Results: Compared with chronic disease patients who chose private medical institutions, patients who chose public medical institutions increased their total cost by 44.9%, total out-of-pocket cost by 22.9%, and decreased the total out-of-pocket ratio by 0.117%, total drug cost out-of-pocket ratio by 0.075%, and drug cost ratio by 0.102%. (4) Conclusions: According to the triple principal-agent relationships, the resource advantages given by the government to public medical institutions, the salary incentive system of medical institutions, and the information asymmetry advantage held by physicians may be important factors for the increase in health costs for chronic disease patients.

## 1. Introduction

The growing health costs are an important issue of health policy in China and the world [1]. Rising health costs and government budget constraints put economic pressure on medical institutions [2]. The huge group of chronic disease patients had multiple cycles of treatment and faced a greater medical burden. This problem was especially evident in China. The medical expenditure of households with chronic disease accounted for about 16.3%, which caused catastrophic medical expenditure in 14% of the households [3].

There were some reasons to expect that the nature of the medical facility may be relevant to health costs. According to the previous research of the comparison between public and private medical institutions, Pavel et al. argued that the total outpatient cost of public medical institutions was always higher than private medical institutions [4]. Excessive drug use in public medical institutions increased the financial burden of patients [5]. There were also scholars who expressed different views. Brugha and Zwi pointed out that the health costs of private medical institutions were higher than that of public medical institutions [6]. Pongsupap and Lerberghe believed that the drug cost of private medical institutions was at least twice that of public medical institutions [7]. Private medical institutions faced the obstacle of patient out-of-pocket costs [8]. Overall, previous research has shown differences in health costs between public and private medical institutions.

Nevertheless, studies on medical choice and chronic disease patients’ health costs left important research gaps. Group-level studies have found that patient attendance at public or private medical institutions affected health costs, but it was unclear whether chronic disease patients were affected by their medical choice. Difficulties in evaluating the effect of medical choice on health costs could also depend on the characteristics of the public and private medical institutions. Institutions’ operational autonomy, the relief of health insurance for patients’ financial pressure, and the patient payment system affecting the patients’ out-of-pocket cost [9,10] should be considered. In addition, previous research had insufficient longitudinal and dynamic tracking of the same individual, which weakened the accuracy of differences between medical institutions of different natures. We viewed the chronic disease patients as existing on a continuum, and changes in individual medical choices could more objectively show the effect of the medical institutions’ nature on health costs. To address this gap in knowledge, we used a quasi-experimental method to set up private medical institutions as the control group and public medical institutions as the experimental group for analysis. We examined the long-term effect of the changes in medical choice on health costs for the same individual. More specifically, we assessed: (1) the changes in health costs in public and private medical institutions from 2011 to 2018; (2) how the medical choice of chronic disease patients affected the health costs; (3) if some patients’ characteristics as living in urban vs. rural areas, being middle-aged vs. elderly, having multiple chronic diseases, and being covered by health insurance were associated with the health costs.

## 2. Theoretical Framework: Principal-Agent Theory

The principal-agent relationship was essentially a contractual relationship, representing the process by which a principal “hires” an agent to perform certain tasks [11]. In previous studies, the principal-agent theory was widely used in the field of medicine, for example, to explain the differences in the efficiency of medical institutions [12] and decision-making processes for healthcare providers [13]. This paper applied the principal-agent theory and argued that there was a triple principal-agent relationship in China’s public and private medical institutions, as shown in Figure 1.

The first principal-agent relationship existed between the government and medical institutions. In China, the government entrusted medical institutions to meet the public health needs. The government was responsible for a series of functions, such as finance and management, and played the role of “principal”. The medical institutions undertook the function of providing a health service and played the role of “agent”. However, this principal-agent relationship differed between public and private medical institutions. The government assumed greater responsibility in the operation of public medical institutions, and private medical institutions were fully responsible for their own profits and losses [14]. In contrast, public medical institutions had a preference for public interests, obviously [15]. There were other problems in public medical institutions. On the one hand, there was the multiple principal problem. Conflicts of interest between different ministries were common, thus creating a cooperation dilemma [9]. Multiple subjects were common in public medical institutions. This complex principal-agent relationship reduced the efficiency of public medical institutions. On the other hand, government subsidies were only a small part of the medical institutions’ operating budget, so the medical institutions must generate revenue [16], which may be contrary to the government’s mandate. There would also be conflicting interests between governments and public medical institutions [17].

The second principal-agent relationship existed between medical institutions and physicians. Physicians were the direct health providers and a key factor in improving the efficiency and effectiveness of hospital operations [18]. Therefore, physicians always undertook the entrustment from two aspects of medical institutions: providing a health service and generating revenue for medical institutions. There were some differences in this relationship between public and private medical institutions. Public medical institutions provided physicians with better job security and benefits than medical institutions [14]. However, there were also studies suggested that public medical institutions required physicians to generate income [19], which could lead to physicians acting in their own best interests, contrary to medical institutions. Physicians in public medical institutions faced financial pressure, and if the incentives were inappropriate, it was difficult to control the moral hazard [20]. Only by improving incentives could the interests of medical institutions and physicians be aligned [21]. Compared with public medical institutions, private medical institutions had better salary management standards [1], and private medical institutions appeared to be more sensitive to incentives than public medical institutions [22].

The third principal-agent relationship existed between the physicians and the patients. From a purely market perspective, the exchanges that typically took place between physicians and patients could be described as information exchanges or transactions [23]. The patients, as the principal, required the physicians (agent) to act in the interests of the patients, and put the responsibility risk on the physicians. Physicians had an advantage in medical knowledge over patients, undertook the agency power of patients’ health [11], and played a unilateral and decisive role in the treatment of patients. Physicians were faced with a situation of a multiple principal problem. The physicians were both the agent of the patient and the agent of the medical institutions [24]. Physicians sought to maximize their own profits at the expense of patients, especially with distorted incentives [25]. In this regard, it seemed that physicians in public medical institutions faced more skepticism. Previous research has suggested that public medical institutions’ physicians relied on bonuses and commissions as part of their normal income, which inevitably led to overtreatment [19]. In this case, the physicians’ interests may differ from the patients’ interests. Because of the organizational bonus system of public medical institutions, for-profit-oriented practices were not uncommon, and physicians’ practices could harm patients’ benefits [26]. In public medical institutions, there was a lack of sufficient market incentives for physicians to take additional initiatives or efforts to improve the patients’ condition [27]. In contrast, private medical institutions were more patient-centric [7], and attracting customers was their goal.

## 3. Data and Methods

### 3.1. Data Source and Variable Identification

Data were drawn from CHARLS (http://charls.pku.edu.cn/, accessed on 3 August 2021), which used a representative sample of people aged 45 years old and above [28]. CHARLS was used to analyze the problem of population aging, and to promote interdisciplinary research on the problem of aging in China. Baseline data collection for CHARLS was implemented in 28 provinces of China in 2011, followed by 3 follow-up interviews in 2013, 2015, and 2018. 

The independent variable was the medical choice between public and private medical institutions of chronic disease patients, and its data came from the answer to “Is this facility public or private?” in the questionnaire. The dependent variables were health costs, and the data came from the questionnaire questions “What was the total cost of this visit, including both treatment and medication cost (includes prescriptions you received)?” and other responses about health costs in the questionnaire. In order to better show the accuracy of the data, this paper performed a logarithmic transformation on the cost items in the regression.

### 3.2. Study Design and Sample Selection

First, the chronic disease patients who participated in the questionnaire survey and utilized a health service in the four-year data were found through ID matching, and the annual data were divided into samples for choosing public medical institutions and private medical institutions. Then, the public medical institutions were taken as the experimental group and individuals who chose private medical institutions in other years were used as the control group. After a longitudinal merger, a total sample of the changes in individual medical choice was formed. Finally, the data of individuals in some years would be overlapped repeatedly. After the duplicate data were deleted from this paper, a total of 1043 samples of 428 participants in the longitudinal survey met the criteria for inclusion in the analysis. See Figure 2 for details.

### 3.3. Methods

Taking private medical institutions as the control group and public medical institutions as the experimental group, this paper explored the effect of individual medical choice on health costs through panel data regression. Given that the outcome variables were continuous variables, the OLS model was used for regression in this study. All data were statistically analyzed using Stata/SE 15.1 software (Stata Corp, College Station, TX, USA).

## 4. Results

### 4.1. Descriptive Statistics

#### 4.1.1. Socio-Demographic Characteristics

Participants’ individual social and health characteristics are summarized in Table 1. Participants’ average age was 58.98 years, 155 (36.21%) of 428 were male and 273 (63.79%) were female, 62 (14.49%) were living in urban area and 364 (85.05%) were living in rural, and the education level at primary school and below was 326 (76.17%). Of the 428 participants, 378 (88.32%) were married, the average income was 8541.52 yuan, 408 (95.33%) had health insurance, and 296 (69.16%) had multiple chronic diseases. 

#### 4.1.2. Changes in Health Costs of Chronic Disease Patients from 2011 to 2018

Table 2 was a comparison of the health costs from 2011 to 2018 (there was no question about the last drug cost in the 2018 questionnaire, so this article only calculated the total cost, total out-of-pocket cost, and out-of-pocket ratio in 2018). From 2011 to 2018, the total cost of chronic disease patients in public medical institutions increased from 413.693 yuan to 1651.921 yuan, with an average of 1005.056 yuan. The total cost in private medical institutions increased from 212.012 yuan to 445.349 yuan, with an average of 290.932 yuan. From 2011 to 2015, the total drug cost in public medical institutions increased from 262.396 yuan to 322.810 yuan, with an average of 306.666 yuan. The total drug cost in private medical institutions decreased from 191.044 yuan to 161.046 yuan, with an average of 166.966 yuan. From 2011 to 2015, the drug cost ratio of chronic disease patients in public medical institutions decreased from 0.763 to 0.646, with an average of 0.716. In private medical institutions, it decreased from 0.873 to 0.845, with an average of 0.833. From 2011 to 2018, the total out-of-pocket cost of chronic disease patients in public medical institutions increased from 360.544 yuan to 1039.871 yuan, with an average of 705.932 yuan. In private medical institutions, it increased from 211.038 yuan to 453.515 yuan, with an average of 272.355 yuan. From 2011 to 2018, the total cost out-of-pocket ratio of chronic disease patients in public medical institutions decreased from 0.925 to 0.843, with an average of 0.859. In private medical institutions, it increased from 0.935 to 0.980, with an average of 0.960. From 2011 to 2015, the out-of-pocket total drug cost of chronic disease patients in public medical institutions decreased from 254.824 yuan to 219.438 yuan, with an average of 259.155 yuan. In private medical institutions, it decreased from 201.422 yuan to 154.201 yuan, with an average of 154.975 yuan. From 2011 to 2015, the total drug cost out-of-pocket ratio of chronic disease patients in public medical institutions decreased from 0.932 to 0.881, with an average of 0.918. In private medical institutions, it decreased from 1.000 to 0.979, with an average of 0.975. See Table 2 for details.

#### 4.1.3. Individual Longitudinal Comparison of Health Costs and Related Ratios

From Table 3, it could be seen that the chronic disease patients who chose public medical institutions in 2011 and chose private medical institutions in 2013 had an increase of 56.230 yuan in the total cost, an increase of 47.125 yuan in the total out-of-pocket cost, an increase of 54.695 yuan in the total drug cost, an increase of 90.638 yuan in the out-of-pocket total drug cost, a decrease of 0.034 in the total cost out-of-pocket ratio, a decrease of 0.041 in the total drug cost out-of-pocket ratio, and an increase of 0.013 in the drug cost ratio. Chronic disease patients who chose public medical institutions in 2013 and chose private medical institutions in 2011 had an increase of 360.099 yuan in the total cost, an increase of 303.422 yuan in the total out-of-pocket cost, an increase of 157.536 yuan in the total drug cost, an increase of 95.767 yuan in the out-of-pocket total drug cost, a decrease of 0.107 in the total cost out-of-pocket ratio, a decrease of 0.069 in the total drug cost out-of-pocket ratio, and a decrease of 0.120 in the drug cost ratio.

### 4.2. Regression Results

#### Main Regression Results

Table 4 shows the estimated results of the effect of medical choice on the health costs of chronic disease patients. The regression results showed that the R^2^ value of the total cost model was 0.236, of the total out-of-pocket cost was 0.196, of the total drug cost was 0.166, of the out-of-pocket total drug cost was 0.171, and that the model’s fit was good. The *p*-values were all less than 0.001, and the model had statistical significance. Compared with patients who chose private medical institutions, chronic disease patients who chose public medical institutions increased their total cost by 44.9% and total out-of-pocket cost by 22.9%.

Table 5 shows the estimated results of the medical choice to the related ratios of the health costs of chronic disease patients. Compared with private medical institutions, the total out-of-pocket ratio of chronic disease patients who chose public medical institutions was reduced by 0.117%, the total drug cost out-of-pocket ratio was reduced by about 0.075%, and the drug cost ratio was reduced by about 0.102%.

### 4.3. Sub-Sample Regression Results

In order to further explore the influence of medical choice on health costs, and considering the influence of different personal characteristics and medical security resources, this paper conducted a sub-sample regression to show the heterogeneity of influence. See Table 6 for details.

#### 4.3.1. Comparison of Chronic Disease Patients Living in Rural and Urban

It could be seen that the choice of public medical institutions for medical treatment increased the total cost of rural chronic disease patients by about 52.7%, the total out-of-pocket cost was increased by 29.3%, the total drug cost was increased by about 28.1%, and the total cost out-of-pocket ratio was reduced by 0.115%. The out-of-pocket ratio was 0.082% less, and the drug cost ratio was 0.093% less. Among urban chronic disease patients, public medical institutions reduced the total cost out-of-pocket ratio by 0.162%, and the drug cost ratio was reduced by 0.319%.

#### 4.3.2. Comparison of Middle-Aged and Elderly Chronic Disease Groups

In the middle-aged group (aged 45 to 64), the total cost of chronic disease patients in public medical institutions was increased by about 44%, and the total out-of-pocket cost was increased by 28.5%, the total cost out-of-pocket ratio was 0.098% less, the total drug cost out-of-pocket ratio was 0.056% less, and the drug cost ratio was 0.084% less. Among the elderly group (aged 65 and older), choosing public medical institutions for medical treatment increased the total cost by about 46.5%, the total cost out-of-pocket ratio was reduced by 0.183%, the total drug cost out-of-pocket ratio was reduced by 0.154%, and the drug cost ratio was reduced by 0.192%.

#### 4.3.3. Comparison of Groups without Multiple Chronic Diseases and Those with Multiple Chronic Diseases

In the group without multiple chronic diseases, the total cost out-of-pocket ratio of the chronic disease patient in public medical institutions was 0.142% less, and the total drug cost out-of-pocket ratio was 0.12% less. Among the groups with multiple chronic diseases, the choice of public medical institutions increased the total cost by about 53.9%, the total out-of-pocket cost was 29.65% higher, the total drug cost was 28.2% higher, the total cost out-of-pocket ratio was less than 0.107%, the total drug cost out-of-pocket ratio was 0.064% less, and the drug cost ratio was 0.11% less.

#### 4.3.4. Comparison of Chronic Disease Patients without Health Insurance and Those with Health Insurance

The results showed that after the change of medical choice, public medical institutions increased the total cost of the without health insurance group by about 132.7%, the total drug cost increased by 124.8%, and the out-of-pocket total drug cost increased by 119.4%. Among chronic disease patients with health insurance, compared with private medical institutions, the total cost of chronic disease patients who chose public medical institutions increased by about 42%, the total out-of-pocket cost was 19.8% higher, the total out-of-pocket ratio was 0.121% less, the total drug cost out-of-pocket ratio was 0.075% less, and the drug cost ratio was 0.099% less.

## 5. Discussion

Our study aimed to discover the effect of medical choice on health costs for chronic disease patients in China. Different from the comparison between groups in previous research, this paper attempted to construct a control group and an experimental group based on the medical choice of individuals through quasi-natural experiments from 2011 to 2018. We have explored the health costs of public and private medical institutions. The results showed that the health costs of chronic disease patients were rising, and the chronic disease patients who chose to seek medical treatment had higher health costs in public medical institutions. The total cost and total out-of-pocket cost in public medical institutions were significantly higher than those in private medical institutions, while the total cost out-of-pocket ratio, total drug cost out-of-pocket ratio, and drug cost ratio in public medical institutions were lower than those of private medical institutions. Further sub-sample regressions examined this outcome and found significance for the total drug cost and out-of-pocket drug cost.

According to the first principal-agent relationship between the government and medical institutions constructed in this paper, the government endowed public medical institutions with more health service functions [29], and public medical institutions usually had more health resources and higher capabilities than those of private medical institutions [30]. Public medical institutions had more total assets and more expensive medical equipment and employed more employees and physicians [31]. This made public medical institutions have the advantages of technical equipment and greater attractiveness. They have gradually shifted from “feeding medical institutions with drug sales” to “feeding medical institutions with medical examinations”, adding unnecessary services or excessive medical treatment [32]. As a result, the total cost and total out-of-pocket cost in public medical institutions may be significantly higher than those of private medical institutions. For chronic diseases, more complex cases [33] or patients with serious health conditions chose public medical institutions [34]. This also led to more spending in public medical institutions due to chronic disease conditions. Previous studies have shown that the purpose of private medical institutions was profit [15], and private medical institutions seemed to have lower health costs. However, Chinese patients showed a high preference for public medical institutions [29], which were still an important concern for chronic disease patients.

According to the second principal-agent relationship, physicians generally paid attention to the compensation obtained from medical institutions [35]. Physicians’ compensation was related to the costs of patients, which was also in line with the logic of previous research. If physicians’ compensation was not related to work effort, they would face a trade-off between patients’ interests and monetary interests [36]. From this point of view, the compensation system that medical institutions gave to physicians may become the key to health costs [19]. Public medical institutions need to generate revenue. Under this requirement, more than 80% of physicians reported overtreatment in the form of prescribing unnecessary diagnoses and tests [37], which also led to higher health costs in public medical institutions. Private medical institutions must show price competitiveness in order to have a competitive advantage, which was also in line with previous findings that private medical institutions had lower prices [34].

The principal-agent relationship between physicians and patients conformed to the information asymmetry in the medical market [38]. Physicians had the right to decide how a medical service was provided, as well as to choose whether and which resources to use [21]. Physicians were likely to provide excessive medical services in order to obtain more profit [39]. After the zero-markup policy for drug sales, physicians in public medical institutions marked up prices for tests and examinations using expensive medical equipment [40]. This resulted in a higher total cost and total out-of-pocket cost for public medical institutions. Chronic disease treatment was expensive [41], and in the control of chronic diseases, the reimbursement ratio for drugs was low [10]. However, more and more patients had higher standard requirements for drugs such as expected import and fast-acting drugs or examinations [42]. Private medical institutions carried more drugs than public medical institutions [43]. In the private medical institutions, there were medicines (such as generic medicines) that were priced higher than the public medical institutions and were generally not reimbursed under health insurance plans [44]. Physicians in private medical institutions were more likely to prescribe drugs that were not covered by the health insurance reimbursement plans. As a result, private medical institutions had a higher total drug cost out-of-pocket ratio. Coupled with the high frequency of use of technical equipment by public medical institutions, the drug cost ratio in public medical institutions was lower than that in private medical institutions.

## 6. Conclusions

This study was the first to estimate the effect of medical choice on health costs for chronic disease patients in China, and it could have scientific and practical implications for the health system reform, since it suggested that the patients’ propension towards public or private medical institutions significantly impacted the health costs. In conclusion, there was increased health costs for chronic disease patients, and the health costs of public medical institutions was higher than that of private medical institutions. In order to alleviate the problem of “expensive health costs”, the government should strengthen the price system control of medical institutions, formulate payment plans for chronic disease patients, and appropriately support the development of private medical institutions. Medical institutions should improve the salary system, reduce the proportion of performance such as examinations and drug costs, and combine the salary of physicians with the social value they create. Physicians should pay more attention to their own medical ethics and establish a patient-centered medical practice.

## 7. Limitations

Several limitations need to be acknowledged. Due to data limitations, on the one hand, the health costs data used in this paper were the costs of chronic disease patients in their most recent visit, but it was not possible to determine whether the health costs were caused by the same chronic disease in the two-year comparison. In order to make up for this defect as much as possible, this paper also selected chronic disease patients who had participated in the questionnaire for four years to ensure that they would visit physicians for the same chronic disease to the greatest extent possible and incur costs. On the other hand, this paper was a quasi-natural experimental study, and the selection of personal characteristics was as consistent as possible, and due to personal decision preferences and other reasons, the patients’ decisions vary each year, and the health costs may be affected.

## Figures and Tables

**Figure 1 ijerph-19-07570-f001:**
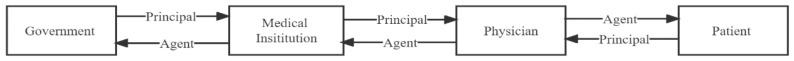
Theoretical framework.

**Figure 2 ijerph-19-07570-f002:**
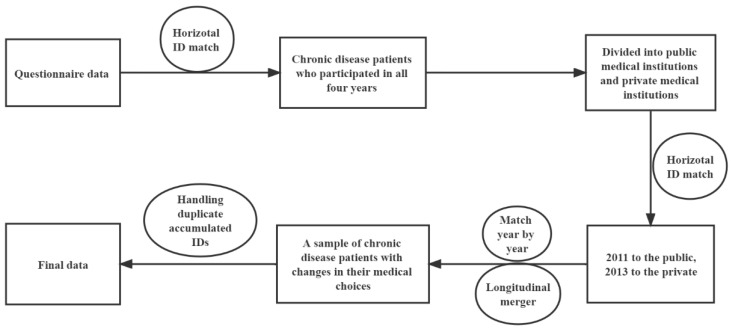
Study design.

**Table 1 ijerph-19-07570-t001:** Basic socio-demographic characteristics (2011, *n* = 428).

Variable	*n*	%
Gender		
Male	155	36.21
Female	273	63.79
Age mean ± S.D.	58.98 ± 8.82	
Living area		
Urban	62	14.49
Rural	364	85.05
Missing value	2	0.47
Education		
Primary school and below	326	76.17
Junior high school	69	16.12
High school and above	33	7.71
Marital status		
Married	378	88.32
Others	50	11.68
Income (yuan) Mean ± S.D.	8541.52 ± 29,723.88	
Missing value	1	0.23
Health insurance		
YES	408	95.33
NO	20	4.67
Multiple chronic diseases		
YES	296	69.16
NO	132	30.84

**Table 2 ijerph-19-07570-t002:** Descriptive statistics of health costs and related ratios.

		Average	2011	2013	2015	2018
Total cost (yuan)	Public	1005.056 [771.140–1238.971]	413.693	672.632	1280.838	1651.921
	*n*	516	127	131	130	128
	Private	290.932 [197.455–384.410]	212.012	240.642	259.261	445.349
	*n*	467	41	161	159	106
Total drug cost (yuan)	Public	306.666 [243.921–369.410]	262.396	335.489	322.810	-
	*n*	315	106	104	105	-
	Private	166.966 [129.244–204.688]	191.044	166.631	161.046	-
	*n*	294	34	129	131	-
Drug cost ratio	Public	0.716 [0.679–0.753]	0.763	0.738	0.646	-
	*n*	311	106	103	102	-
	Private	0.833 [0.799–0.867]	0.873	0.810	0.845	-
	*n*	293	34	129	130	-
Total out-of-pocket cost (yuan)	Public	705.932 [534.285–877.579]	360.544	539.845	855.813	1039.871
	*n*	515	125	127	128	135
	Private	272.355 [179.618–365.063]	211.038	209.692	232.695	453.515
	*n*	460	40	158	159	103
Total cost out-of-pocket ratio	Public	0.859 [0.836–0.881]	0.925	0.861	0.804	0.843
	*n*	497	125	125	121	126
	Private	0.960 [0.945–0.975]	0.935	0.942	0.971	0.980
	*n*	456	40	158	156	102
Out-of-pocket total drug cost (yuan)	Public	259.155 [209.156–309.153]	254.824	305.740	219.438	-
	*n*	306	102	99	105	-
	Private	154.975 [120.555–189.366]	201.422	143.643	154.201	-
	*n*	288	32	122	134	-
Total drug cost out-of-pocket ratio	Public	0.918 [0.894–0.942]	0.932	0.941	0.881	-
	*n*	296	101	97	98	-
	Private	0.975 [0.961–0.990]	1.000	0.965	0.979	-
	*n*	276	30	121	125	-

Note: 95%CI in square brackets.

**Table 3 ijerph-19-07570-t003:** Changes in health costs for medical choice.

	Changes	Total Cost (Yuan)	Total Out-of-Pocket Cost (Yuan)	Total Drug Cost (Yuan)	Out-of-Pocket Total Drug Cost (Yuan)	Total Cost Out-of-Pocket Ratio	Total Drug Cost Out-of-Pocket Ratio	Drug Cost Ratio
Medical Choice								
Public (2011)–Private (2013)		56.230	47.125	54.695	90.638	−0.034	−0.041	0.013
Public (2011)–Private (2015)		177.354	118.911	72.899	78.165	−0.076	−0.071	−0.126
Public (2011)–Private (2018)		183.030	158.941	-	-	0.013	-	-
Public (2013)–Private (2011)		360.099	303.422	157.536	95.767	−0.107	−0.069	−0.120
Public (2013)–Private (2015)		306.265	183.805	119.794	102.644	−0.104	−0.043	0.108
Public (2013)–Private (2018)		541.023	422.455	-	-	−0.074	-	-
Public (2015)–Private (2011)		1606.158	1542.174	−126.072	−148.033	−0.075	−0.201	−0.161
Public (2015)–Private (2013)		1190.209	562.603	240.822	138.813	−0.174	−0.066	−0.212
Public (2015)–Private (2018)		61.335	−70.276	-	-	−0.125	-	-
Public (2018)–Private (2011)		942.424	506.500	-	-	−0.179	-	-
Public (2018)–Private (2013)		1192.337	617.690	-	-	−0.129	-	-
Public (2018)–Private (2015)		1490.438	986.098	-	-	−0.073	-	-

Note: “Public” means choosing public medical institution; “Private” means choosing private medical institution. In parentheses is the year the patient chose the medical institution. The data in the table are the changes.

**Table 4 ijerph-19-07570-t004:** The effect of medical choice on health costs amounts.

	Total Cost (Yuan)	Total Out-of-Pocket Cost (Yuan)	Total Drug Cost (Yuan)	Out-of-Pocket Total Drug Cost (Yuan)
Medical choice	0.449 ***	0.229 **	0.213	0.077
	(0.097)	(0.107)	(0.137)	(0.133)
Year (Ref: 2011)				
2013	0.326 **	0.273 *	0.165	0.081
	(0.133)	(0.143)	(0.168)	(0.161)
2015	0.534 ***	0.433 ***	0.271 *	0.154
	(0.139)	(0.147)	(0.161)	(0.153)
2018	0.662 ***	0.671 ***	-	-
	(0.143)	(0.145)	-	-
Gender (Ref: Female)	0.171 *	0.092	0.422 ***	0.311 **
	(0.101)	(0.109)	(0.125)	(0.128)
Living area (Ref: Rural)	0.282 **	0.204	0.085	0.012
	(0.127)	(0.151)	(0.183)	(0.196)
Age	0.004	−0.002	0.002	−0.000
	(0.006)	(0.006)	(0.008)	(0.008)
Marital status (Ref: Others)	−0.007	0.051	−0.077	−0.109
	(0.163)	(0.168)	(0.186)	(0.189)
Education (Ref: Primary school and below)				
Junior high school	0.003	0.045	0.030	0.026
	(0.151)	(0.158)	(0.183)	(0.193)
High school and above	−0.099	−0.223	−0.030	−0.016
	(0.177)	(0.190)	(0.187)	(0.193)
Income	0.002	−0.012	0.008	0.002
	(0.012)	(0.012)	(0.015)	(0.016)
Institutional level (Ref: Primary medical institutions)	1.335 ***	1.342 ***	1.206 ***	1.326 ***
	(0.116)	(0.116)	(0.163)	(0.157)
Health insurance (Ref: Not have)	−0.242	−0.158	−0.269	−0.285
	(0.212)	(0.234)	(0.222)	(0.223)
Multiple chronic diseases (Ref: No)	0.020	0.087	−0.048	−0.092
	(0.106)	(0.111)	(0.135)	(0.144)
Constant term	3.879 ***	4.153 ***	3.866 ***	4.209 ***
	(0.466)	(0.504)	(0.573)	(0.599)
R^2^	0.236	0.196	0.166	0.171
*p*	<0.001	<0.001	<0.001	<0.001
*n*	975	969	605	586

Note: Robust standard errors in brackets, * *p* < 0.1, ** *p* < 0.05, *** *p* < 0.01.

**Table 5 ijerph-19-07570-t005:** The effect of medical choice on health costs-related ratios.

	Total Cost Out-of-Pocket Ratio	Total Drug Cost Out-of-Pocket Ratio	Drug Cost Ratio
Medical choice	−0.117 ***	−0.075 ***	−0.102 ***
	(0.017)	(0.016)	(0.029)
Year (Ref: 2011)			
2013	−0.055 **	−0.008	−0.069 *
	(0.022)	(0.020)	(0.036)
2015	−0.066 ***	−0.027	−0.092 ***
	(0.023)	(0.022)	(0.034)
2018	−0.048 **	-	-
	(0.022)	-	-
Gender (Ref: Female)	−0.011	0.004	0.041
	(0.017)	(0.019)	(0.026)
Living area (Ref: Rural)	−0.029	−0.004	−0.090 **
	(0.025)	(0.027)	(0.042)
Age	−0.002 **	−0.002 *	−0.001
	(0.001)	(0.001)	(0.002)
Marital status (Ref: Others)	−0.008	−0.028	0.051
	(0.020)	(0.025)	(0.039)
Education (Ref: Primary school and below)			
Junior high school	−0.008	−0.028	0.051
	(0.020)	(0.025)	(0.039)
High school and above	−0.034	0.027	0.096 *
	(0.035)	(0.017)	(0.052)
Income	−0.001	−0.000	−0.003
	(0.002)	(0.002)	(0.003)
Institutional level (Ref: Primary medical institutions)	0.017	0.040 **	−0.108 ***
	(0.020)	(0.019)	(0.034)
Health insurance (Ref: Not have)	−0.021	−0.027	−0.054
	(0.032)	(0.023)	(0.045)
Multiple chronic diseases (Ref: No)	0.017	0.012	−0.038
	(0.016)	(0.019)	(0.029)
Constant term	1.159 ***	1.102 ***	1.066 ***
	(0.080)	(0.080)	(0.124)
R^2^	0.073	0.054	0.086
*p*	<0.001	<0.01	<0.001
*n*	947	570	600

Note: Robust standard errors in brackets, * *p* < 0.1, ** *p* < 0.05, *** *p* < 0.01.

**Table 6 ijerph-19-07570-t006:** Estimated results of the effect of medical choice on health costs.

	Rural	Urban	Middle- Aged	Elderly	No Multiple Chronic Diseases	Multiple Chronic Diseases	No Health Insurance	Health Insurance
Total cost	0.527 ***	−0.077	0.440 ***	0.465 ***	0.249	0.539 ***	1.327 ***	0.420 ***
	(0.100)	(0.341)	(0.114)	(0.178)	(0.174)	(0.117)	(0.476)	(0.099)
Total out-of-pocket cost	0.293 ***	−0.211	0.285 **	0.003	0.075	0.295 **	0.896	0.198 *
	(0.110)	(0.413)	(0.123)	(0.211)	(0.229)	(0.117)	(0.629)	(0.109)
Total drug cost	0.281 **	−0.220	0.251	0.071	0.101	0.282 *	1.248 **	0.170
	(0.142)	(0.574)	(0.159)	(0.243)	(0.245)	(0.167)	(0.528)	(0.140)
Out-of-pocket total drug cost	0.113	−0.038	0.116	−0.053	0.009	0.081	1.194 **	0.029
	(0.140)	(0.513)	(0.153)	(0.270)	(0.256)	(0.160)	(0.528)	(0.137)
Total cost out-of-pocket ratio	−0.115 ***	−0.162 **	−0.098 ***	−0.183 ***	−0.142 ***	−0.107 ***	−0.061	−0.121 ***
	(0.017)	(0.068)	(0.019)	(0.036)	(0.038)	(0.018)	(0.051)	(0.017)
Total drug cost out-of-pocket ratio	−0.082 ***	−0.029	−0.056 ***	−0.154 ***	−0.120 ***	−0.064 ***	−0.021	−0.075 ***
	(0.017)	(0.035)	(0.015)	(0.050)	(0.042)	(0.017)	(0.031)	(0.016)
Drug cost ratio	−0.093 ***	−0.319 ***	−0.084 **	−0.192 ***	−0.078	−0.110 ***	−0.121	−0.099 ***
	(0.029)	(0.114)	(0.033)	(0.057)	(0.053)	(0.035)	(0.084)	(0.030)

Note: Robust standard errors in brackets, * *p* < 0.1, ** *p* < 0.05, *** *p* < 0.01. For the sake of brevity, Table 6 does not report the regression results of the remaining explanatory variables and constant terms. The regression results for each column are the effect of medical choice on health costs in different sub-samples.

## Data Availability

Publicly available datasets were analyzed in this study. This data can be found at: http://charls.pku.edu.cn (accessed on 3 August 2021).

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
