# Peer review of "The Effect of Medical Choice on Health Costs of Middle-Aged and Elderly Patients with Chronic Disease: Based on Principal-Agent Theory"

_ijerph, 2022, doi:10.3390/ijerph19137570_

Round 1
Reviewer 1 Report
In this article, Li D. and colleagues have investigated the changes in health cost and the effects of medical choice evaluating a population of chronic disease patients in China, between 2011 and 2018.
The study is properly designed, yet the framework in Figure 1 seems to be discordant with its description in the text and the temporal circumstances supporting the introduction could be better defined.
The results may be illustrated in a more clear and linear arrangement, especially in Table 3, and the content of Table 4 and Table 5 should be better explained in the respective captions.
In my opinion, this manuscript could be improved within the exposition of results and conclusion, and a revision of the english language.
Author Response
Point 1: The study is properly designed, yet the framework in Figure 1 seems to be discordant with its description in the text and the temporal circumstances supporting the introduction could be better defined.
Response 1: Thank you for your recognition of our research design, and thanks for your constructive comment. We have revised the description of the relationship between physicians and patients in Figure 1, and have further enriched the Theoretical Framework. We have further defined the temporal circumstances supporting the introduction.
It has been revised in the manuscript:
In the first principal-agent relationship, we have mainly added:
There were other problems in public medical institutions. On the one hand, there was the problem of the multiple principal problem. Conflicts of interest between different ministries were not uncommon and therefore did not actively cooperate [1]. Multiple subjects were common in public medical institutions. This complex principal-agent relationship reduced the efficiency of public medical institutions. On the other hand, government subsidies were only a small part of the medical institutions’ operating budget, so the medical institutions must generate revenue, which may be contrary to the government’s mandate [2]. There would also be conflicting interests between governments and public medical institutions [3].
In the second principal-agent relationship, we have mainly added:
Public medical institutions required physicians to generate income [4], which could lead to physicians acting in their own best interests contrary to medical institutions. Only by improving incentives could the interests of medical institutions and physicians be aligned [5]. Compared with public medical institutions, private medical institutions had better salary management standards [6], and private medical institutions appeared to be more sensitive to incentives than public medical institutions [7].
The third principal-agent relationship, we have mainly added:
Physicians faced with a situation of multiple principal problem. The physicians were both the agent of the patient and the agent of the medical institution [8]. Physicians sought to maximize their own profits at the expense of patients, especially with distorted incentives [9].
In public medical institutions, there was a lack of sufficient market incentives for physicians to take additional initiatives or efforts to improve the patients’ condition [10]. In contrast, Private medical institutions were more patient-centric [11], and attracting customers was their goal.
Reference:
[1] Wang Y, Castelli A, Cao Q, et al. Assessing the design of China’s complex health system–Concerns on equity and efficiency. Health Policy Open2020, 1,100021. https://doi.org/10.1016/j.hpopen.2020.100021.
[2] He A J, Qian J. Explaining medical disputes in Chinese public hospitals: the doctor–patient relationship and its implications for health policy reforms. Health Economics, Policy and Law2016, 11(4), 359-378. https://doi.org/10.1017/S1744133116000128
[3] Zhang H, Hu H, Wu C, et al. Impact of China’s public hospital reform on healthcare expenditures and utilization: a case study in ZJ Province. PLoS One2015, 10(11), e0143130. https://doi.org/10.1371/journal.pone.0143130
[4] Chan, C. S. C. Mistrust of physicians in China: society, institution, and interaction as root causes. Developing World Bioeth-ics2018, 18(1), 16-25. https://doi.org/10.1111/dewb.12162
[5] Chmielewska M, Stokwiszewski J, Filip J, et al. Motivation factors affecting the job attitude of medical doctors and the organizational performance of public hospitals in Warsaw, Poland. BMC Health Services Research2020, 20(1),1-12. https://doi.org/10.1186/s12913-020-05573-z
[6] Jing, R.; Xu, T.; Lai, X.; Mahmoudi, E.; Fang, H. Technical efficiency of public and private hospitals in Beijing, China: a comparative study. International journal of environmental research and public health2020, 17(1), 82. https://doi.org/10.3390/ijerph17010082
[7] Kruse F M, Stadhouders N W, Adang E M, et al. Do private hospitals outperform public hospitals regarding efficiency, accessibility, and quality of care in the European Union? A literature review. The International journal of health planning and man-agement2018, 33(2), e434-e453. https://doi.org/10.1002/hpm.2502
[8] Zang W, Zhou M, Zhao S. Deregulation and pricing of medical services: a policy experiment based in China. BMC Health Services Research2021, 21(1), 1-10. https://doi.org/10.1186/s12913-021-06525-x
[9] Nguyen H. The principal-agent problems in health care: evidence from prescribing patterns of private providers in Vietnam. Health policy and planning2011, 26(suppl_1), i53-i62. https://doi.org/10.1093/heapol/czr028
[10] Andaleeb, S. S. Public and private hospitals in Bangladesh: service quality and predictors of hospital choice. Health policy and planning2020, 15(1), 95-102. https://doi.org/10.1093/heapol/15.1.95
[11] Pongsupap, Y.; Lerberghe, W.V. Choosing between public and private or between hospital and primary care: responsiveness, patient-centredness and prescribing patterns in outpatient consultations in Bangkok. Tropical medicine & international health2006, 11(1), 81-89. https://doi.org/10.1111/j.1365-3156.2005.01532.x
Point 2: The results may be illustrated in a more clear and linear arrangement, especially in Table 3, and the content of Table 4 and Table 5 should be better explained in the respective captions.
Response 2: Thanks for your comments. We have restructured table 3, improved the footnotes, and rephrased the results, hopefully making the results look more clear and linear arrangement. Table 4 was mainly used to analyze the effect of medical choice on health cost amounts (total cost, total out-of-pocket cost, total drug cost, out-of-pocket total drug cost). Table 5 was mainly used to analyze the effect of medical choice on health cost related ratios (total cost out-of-pocket ratio, total drug cost out-of-pocket, drug cost ratio).
We have corrected the respective captions of Table 4 and Table 5.
It has been revised in the manuscript:
The caption of Table 4 was revised to “The effect of medical choice on health cost amounts”; “The caption of Table 5 was revised to “The effect of medical choice on health cost related ratios”.
Point 3: In my opinion, this manuscript could be improved within the exposition of results and conclusion, and a revision of the English language.
Response 3: Thanks for your constructive comments. We have revised the grammatical errors and inconsistencies. We have revised the whole manuscript to make it more succinct and to provide clarity. We have significantly revised the result section to make it clearer, and we have edited the conclusion part.
Reviewer 2 Report
the authors presented a well written report focusing on the most important factors for the increase of health cost for chronic disease patients. The subject is really interesting and the authors made the discussion easy to read and to interpret.
Author Response
Takeaway: The authors present a well-written report that focuses on the most important factors in increasing health costs for people with chronic conditions. The topic is very interesting and the author makes the discussion easy to read and explain.
Response: Thanks for your comment. Thank you for acknowledging this research.
Reviewer 3 Report
Thank you for your submission on the interesting research topic of the healthcare cost and its determinants. I’ve also appreciated the use of the Principal Agent theory framework.
Notwithstanding the foregoing, the paper entitled « The Effect of Medical Choice on Health Cost of Middle-Aged and Elderly Patients with Chronic Disease: Based on Principal Agent Theory » shows several issues that require your attention and addressing before the paper can be accepted in my general evaluation.
INTRODUCTION:
- extensive editing of English language and style is required: sentences are too long and some terms (e.g. normal families, chronic disease families) sound bad……in my opinion
- the paper will benefit from an early discussion about the organization of the Chinese health service (public and private medical institutions and insurance programs) and the system of payment by the patients. This is critically important to alert readers to the specific details of the paper. As an example, you could refer to this reference (or others, if you prefer):
Yuxi Wang, Adriana Castelli, Qi Cao, Dan Liu. Assessing the design of China’s complex health system – Concerns on equity and efficiency. https://doi.org/10.1016/j.hpopen.2020.100021
Theoretical framework….
The Principal Agent theory is a very interesting approach but, as you say at the line 83, the principal agent relationship differed between public and private medical institutions. This is the focus of your work, so I think you should further discuss the dilemma existing when an agent is motivated to act in his own best interests, which is contrary to that of his principal (e.g. Medical institutions vs Government or Physicians vs Medical institutions). Another question is the use of performance measures in healthcare, to align the interests of the agent with those of the principal. Finally, the multiple principal problem (when one agent acts on behalf of multiple principals) could be considered because principal-agent problem is intensified. E.g., asymmetric information exists between the principals and agent potentially leading to moral hazard, but there is also asymmetric information between the principals themselves. Multiple principal problem is particularly significant in the public sector, where multiple principals are common and both efficiency and democratic accountability could be weakened by the lack of a clear governance.
Some aspects are reported in the discussion, but they should be introduced earlier.
Data and methods
- Specify, already in the data source section, that your data come from a population survey and data are collected trough paper questionnnaires
- In the study design section I don’t understand the meaning of 1043 pieces. I suppose it refers to the longitudinal nature of the data……
Results
- too many decimal places (eg. change the mean age from 58.979 to to 58.98 years)
- add 95%CI to the average column in Table 2
- In section 4.1.3, between line 188 and 197, I suggest to rephrase the statements about the comparison between 2011 and 2013 beahaviour.
E.g. s “in 2013 had a total cost of 188 56.230 yuan more” could be “had an increase of 56.230 yuan in the total cost”
- In section 4.2.1., line 206, the term « respectively » should refer to something. I understand that it refers to the cost variables (Total cost, Total out-of-pocket cost, Total drug cost, Out-of-pocket total drug cost), but it is not so clear.
- In section 4.3 sub-sample regressione results, the term « group regression » is unclear. You aim to compare groups of patients (rural vs urban, elderly vs middle-aged etc.), but I have some difficulties to understand the regression models shown in Table 6.
- Furthermore, if the 428 participants in the survey have an average age of 58.98 years (± 8.82), when you refer to the « elderly » group, what do you mean ?
Discussion
Discussion must be improved and better organized. I try to suggest some lines of development…
Line 271 : « The total cost and total out-of-pocket cost in public medical institutions were significantly higher than those in private medical institutions. »
Line 280 : « For chronic disease, more complex cases [20] or patients with serious health conditions chose public medical institutions »
Line 284 : « However, Chinese patients showed a high preference for public medical institutions [18], which were still an important concern for chronic disease patients ».
While reading these statements, I immediatly think about the different technological equipment between public and private medical institution. Could it be a determinant of the patients’s choice in China?
Line 272 : « total drug cost out-of-pocket ratio, and drug cost ratio in public medical institutions was lower than that of private medical institutions »
Which is the system of drug prescription in the private medical institutions ? Is it different fom that in the public ?
Finally…..
I hope that my suggestions would be helpful and I sincerely believe the audiences will benefit greatly from the authors' careful proof

Round 2
Reviewer 3 Report
Dear authors, after the revision process the paper was substantially improved but you still need to closely read through the paper for language issues, consistency etc., in particular along the discussion section.
Below, I try to suggest you some minor changes.
Line 46: in my opinion you could substiute comma with dot before Brugha
Line 59 : before “In addiction” : you could move here the statement, currently at the line 71:
“The paper will benefit from early discussions about the public medical institutions and private medical institutions had different operational autonomy, and there were differences in the relief of medical insurances for patients' financial pressure, and the patient payment system would also affect the patient's out-of-pocket cost [9,10]. All these factors may affect the health cost of patients »
After having change it in something like this :
« Difficulties in evaluating the effect of medical choice on healthcare costs could also depend on the characteristics of the public and private medical institutions. Institutions' operational autonomy, the relief of medical insurances for patients' financial pressure and the patient payment system affecting the patient's out-of-pocket cost [9,10] should be considered. »
Line 67: why do you use “to adressed” instead of “to adress”?
Line 75 :
« More specifically, we assessed: 1) The changes in health cost in public and private medical institutions from 2011 to 2018. 2) How the medical choice of chronic disease patients affected health cost. 3) Was this association consistent between urban and rural areas, middle-aged and elderly, patients with multiple chronic diseases and those without multiple chronic diseases, and groups with and without health insurance? »
I suggest to change in:
« More specifically, we assessed: 1) The changes in health cost in public and private medical institutions from 2011 to 2018. 2) How the medical choice of chronic disease patients affected health cost. 3) if some patient’s characterisctics as living in urban vs rural areas, being middle-aged vs elderly, having multiple chronic diseases and being covered by a health insurance are associated with health cost»
Line 85 : why do you use “to explained » instesd of « to explain » ?
Line 268: Compared with patients WHO chose private medical institutions, chronic disease patients WHO chose (or CHOOSING) public medical institutions increased their total cost by 44.9% and 269 total out-of-pocket cost by 22.9%.
Line 328 : Our study aimed to discover the effect of medical choice on health cost for chronic disease patients IN CHINA (I think it’s better to specify, being the first study about this topic in China)
Line 336 : « medical institutions. And total cost out-of-pocket….. » I suggest to rephrase in « medical institutions, while the total cost out-of-pocket…… » (remove And)
Line 356 : under the influence? (not influenced , I think)
Line 360 : According to the second principal-agent relationship constructed in this paper, medical institutions gave physicians remuneration, which was also widely concerned by physicians
(The meaning is not clear….)
Line 406 : I suggest to change
« This study was the first one to estimate the effect of medical choice on health cost for chronic disease patients, the results of which had significantly scientific and practical implications for health system reform. This study added to the evidence suggesting that the health cost was related to patient's choice of medical institution in China ».
in
« This study was the first one to estimate the effect of medical choice on health cost for chronic disease patients in China and it could have scientific and practical implications for the health system reform, since it suggests that the patient's propension towards public or private medical institution significantly impacts the healthcare costs. »
